# Hippo Signaling at the Hallmarks of Cancer and Drug Resistance

**DOI:** 10.3390/cells13070564

**Published:** 2024-03-22

**Authors:** Ramesh Kumar, Wanjin Hong

**Affiliations:** Institute of Molecular and Cell Biology, A*STAR (Agency for Science, Technology, and Research), Singapore 138673, Singapore; mcbhwj@imcb.a-star.edu.sg

**Keywords:** hippo signaling, carcinogenesis, cancer, KRAS, EGFR, drug resistance, combination therapy

## Abstract

Originally identified in *Drosophila melanogaster* in 1995, the Hippo signaling pathway plays a pivotal role in organ size control and tumor suppression by inhibiting proliferation and promoting apoptosis. Large tumor suppressors 1 and 2 (LATS1/2) directly phosphorylate the Yki orthologs YAP (yes-associated protein) and its paralog TAZ (also known as WW domain-containing transcription regulator 1 [WWTR1]), thereby inhibiting their nuclear localization and pairing with transcriptional coactivators TEAD1-4. Earnest efforts from many research laboratories have established the role of mis-regulated Hippo signaling in tumorigenesis, epithelial mesenchymal transition (EMT), oncogenic stemness, and, more recently, development of drug resistances. Hippo signaling components at the heart of oncogenic adaptations fuel the development of drug resistance in many cancers for targeted therapies including KRAS and EGFR mutants. The first U.S. food and drug administration (US FDA) approval of the imatinib tyrosine kinase inhibitor in 2001 paved the way for nearly 100 small-molecule anti-cancer drugs approved by the US FDA and the national medical products administration (NMPA). However, the low response rate and development of drug resistance have posed a major hurdle to improving the progression-free survival (PFS) and overall survival (OS) of cancer patients. Accumulating evidence has enabled scientists and clinicians to strategize the therapeutic approaches of targeting cancer cells and to navigate the development of drug resistance through the continuous monitoring of tumor evolution and oncogenic adaptations. In this review, we highlight the emerging aspects of Hippo signaling in cross-talk with other oncogenic drivers and how this information can be translated into combination therapy to target a broad range of aggressive tumors and the development of drug resistance.

## 1. Introduction

Originally identified and described in *Drosophila melanogaster*, principal components of the mammalian Hippo signaling pathway have been conserved. During the past 15 years, the area has received a wider investigation due to key regulatory roles in development, regeneration, as well as pro-cancerous functions. Key components of the mammalian Hippo signaling pathway consist of mammalian Ste20-like kinases 1/2 (MST1/2), large tumor suppressor 1/2 (LATS1/2), yes-associated protein (YAP), and/or its paralog transcriptional coactivator with a PDZ-binding motif (TAZ) encoded by the WW domain-containing transcription regulator 1 (WWTR1) gene. Upon physiological and/or pathological signal-induced activation of the upstream components, the MST1/2 (Hippo/Hpo) forms a complex with the adaptor protein Salvador 1 (SAV1), which acts to phosphorylate and activate the large tumor suppressor (LATS1/2) and its binding partners MOB kinase activator 1A/B (MOB1A/B) [1]. A dynamic cross-talk between these signaling components leads to the phosphorylation, cytoplasmic retention, and proteasomal degradation of YAP and TAZ proteins. Conversely, the deactivation of the core kinases leads to YAP and TAZ nuclear retention and TEA domain (TEAD) protein binding. The nuclear YAP/TAZ-TEAD transcriptional program regulates the expression of Hippo pathway target genes, governing a broad range of cellular functions, and it is a key determinant of cell growth and proliferations [2].

Post-translational modifications (PTMs) at the heart of the Hippo signaling pathway are key determinants of functional outcomes. YAP/TAZ activity is tightly modulated by the phosphorylation of the serine/threonine (S/T) residues. Out of five known phosphorylated serine residues of YAP by LATS kinases, the “*S127*” of YAP (corresponding S89 of TAZ) has been well investigated and implicated for its key role in mediating 14-3-3 protein binding and subsequent cytoplasmic retention. The phosphorylation by LATS kinases promotes their cytoplasmic localization with YAP (Ser 127 YAP1 and Ser 89 of the TAZ protein) by creating a binding site for 14-3-3 proteins [3]. Tyrosine kinases c-Src and YES1 phosphorylate the Y357 of YAP, leading to the activation of anti-apoptotic and regenerative genes and subsequent transformation. Several context-specific roles of YAP have been reported and reviewed [4]. YAP/TAZ interacts with transcription factors SMAD1, SMAD2/3, and SMAD7; RUNT-related transcription factors (RUNX1 and RUNX2); and T-box transcription factor 5 (TBX5) and p73 [5]. Mathematical models of the Hippo and TGF-β cross-talk indicate that TAZ/YAP can modulate TGF-β receptor activity and promote nuclear retention [6]. Under stress, key energy sensor molecule AMP-activated protein kinase (AMPK) directly phosphorylates and inhibits YAP and TAZ via preventing their interaction with TEAD [7]. There is a positive feedback loop between YAP/TAZ and Notch signaling. YAP/TAZ promotes Notch signaling transduction by activating Jagged1 (JAG1) expression [8]. YAP/TAZ has been found as a component of a destructive complex in the Wnt/β-catenin pathway, enters the nucleus, and forms a complex with accumulated β-catenin to promote the transcriptional activation of a group of target genes such as SRY (sex-determining region Y)-box 2 (SOX2), zinc finger protein SNAI2, Bcl-2-like protein 1(BCL2L1), and Survivin, also known as baculoviral inhibitor of apoptosis repeat-containing 5 (BIRC5), cooperatively inducing cell proliferation and promoting tumorigenesis [9,10].

The transcriptional enhanced associated domain proteins (TEADs) have been well established key partners of YAP/TAZ activity. TEAD contains a “TEA” DNA binding domain that binds to promoters or enhancers of target genes, and a YAP/TAZ binding domain that associates with transcriptional cofactors. Besides N-terminal phosphorylation, C-terminal palmitoylation at the consensus Cystine residues has been shown to modulate TEADs’ stability and the transcriptional activity of target genes [11]. Key cysteine residues C359 in TEAD1, C380 in TEAD2, C371 in TEAD3, and C360 in TEAD4 play very crucial roles in TEAD palmitoylation [12]. Stress -induced p38 MAPK (mitogen-activated protein kinase) has been shown to play a key role in TEADs nucleocytoplasmic shuttling [13]. Vestigial-like family member 4 (VGLL4) inhibits YAP by competing with YAP for its TEAD-binding. T cell lymphoma invasion and metastasis 1 (TIAM1) interacts with YAP/TAZ and impairs the YAP/TAZ-TEAD interaction [14]. Upon the downregulation of the Hippo pathway, YAP/TAZ’s nuclear translocation and binding to TEADs activate the transcription of its target genes including connective tissue growth factor (CTGF), cysteine rich protein 61 (CYR61), and anexelekto (AXL), thus regulating cell cycle and proliferation [15].

Investigations in the past decade have highlighted the roles of Hippo signaling in regenerative and pro-cancerous functions. The regenerative activities have been reported and discussed in great detail [16]. The Hippo signaling pathway is mis-regulated in a wide range of cancers despite the low mutation frequency of the core components within the pathway. The genetic alteration of the Hippo pathway has been increasingly detected in many cancers including glioma, breast, liver, lung, prostate, colorectal, and gastric cancers, with a likely involvement in others because of non-genetic alterations [17]. Components of the core kinases are inactivated via infrequent mutation, typically present in <10% of cancer cases [18]. However, the hyperactivity of YAP/TAZ triggers the onset and metastasis of multiple types of cancers. Animal studies including genetically engineered mouse (GEM) models demonstrate YAP/TAZ as bona fide oncogenic drivers [19]. For example, YAP overexpression or MST1/2 knockout in the liver induces spontaneous liver cancer. Likewise, the ectopic expression of YAP promotes the tumor formation and progression of lung cancer [14]. Conversely, the genetic inactivation of YAP reduces or suppresses tumor formation in various genetically engineered mouse models of lung, breast, colon, and pancreatic cancers [20]. In this review, we present the key oncogenic functions of the TEAD transcription factor and how the YAP/TAZ-TEAD nuclear binding orchestrates the transcriptional program, cellular proliferation, and subsequent tumorigenesis. In the second part, we highlight the cross-talk between Hippo signaling components with two highly prevalent oncogenic mutations of KRAS and EGFR. At the end of the review, we discuss how drugging the Hippo arm in combination with clinical molecules can be a viable option to overcome the oncogenic adaptations and development of drug resistance.

## 2. YAP/TAZ Activation in Multiple Cancers

The YAP and TAZ proteins are well known for their key role in driving cancer cell proliferation, migration, metastasis, and development of drug resistance. YAP/TAZ drives the transcription of key cell cycle genes, such as the cell cycle transcription factor forkhead box protein M1 (*FOXM1*) and its target *CCND1* (encoding cyclin D1) in malignant mesothelioma cells. Data from over 9000 tumor samples indicate that YAP and TAZ are frequently amplified in head and neck and gynecologic cancers [17]. Consistently, the highest amplification frequency of these two oncogenes were revealed in head and neck squamous cell carcinoma (HNSC) and cervical squamous cell carcinoma (CESC), predicted through the Cancer Genome Atlas of 19 Hippo core genes [18]. A high expression of YAP and GLUT3 in different human cancers are positively correlated. YAP may lead to cancer by stimulating glucose uptake and glycolysis by upregulating the expression of Glucose transporter 3 (GLUT3) and subsequent glucose metabolism [21]. A growing body of evidence suggests that “Warburg effect driven” YAP/TAZ activation provides a metabolic competitive advantage and fuels the process of cancer cell survival and maintenance [22]. In cancer cells, YAP/TAZ activity is associated with pro-survival programs and the development of drug resistance. In response to nutrient deprivation, YAP mediates autophagy activation and survival in breast cancer cells, likely due to YAP-induced target gene expression [23]. Collectively, YAP/TAZ-mediated metabolic activities are key regulators of tumorigenesis. Accumulating evidence suggests that high YAP activity is linked to the development of drug resistance in many types of cancer cells. YAP indirectly induces taxol resistance by enhancing the target (CTGF and CYR61) gene expression, which leads to the inhibition of taxol-induced apoptosis by upregulating the anti-apoptotic proteins, B-cell lymphoma-extra-large (Bcl-xL), and clAP1, and promoting cancer cell survival. Likewise, poly (ADP-ribose) polymerase 1 (PARP1) induces stemness and taxol resistance by promoting YAP dephosphorylation and its nuclear translocation in gastric cancer cells. Increased YAP expression was found in 5-FU-resistant colon cancer liver metastases with shorter survival. YAP increases EGFR expression in oesophageal carcinoma and develops resistance to 5-FU and docetaxel. In addition to many DNA damaging agents, YAP mediates resistance to RAF and MEK inhibition in many cancers. Interestingly, YAP depletion in many resistant cancer cell lines restored inhibitors’ sensitivity [24].

### 2.1. YAP/TAZ Activation in EMT and Oncogenic Stemness

Epithelial-to-mesenchymal transition (EMT) is a critical step in most types of early oncogenesis whereby epithelial cells lose their characteristics of cell polarity and cell–cell adhesion and gain mesenchymal cell properties associated with cell motility. EMT is the first event for metastasis where the detachment of cells from the tissue mass takes place. The elevated YAP expression promotes EMT by upregulating the SNAI2 gene (SLUG) expression in colorectal cancer cells [25]. The overexpression of YAP/TAZ induces EMT, which is mostly reversed upon YAP/TAZ inhibition. YAP/TAZ targets genes Axl, Cyr61, and CTGF have been shown to have the ability to induce EMT and stemness. Moreover, YAP has been demonstrated to interact with Smad2/3/4 and regulate the mRNA expression of EMT-inducing transcription factors, Snail, Twist1, and Slug. EMT-inducing transcription factor ZEB1 also interacts with YAP and enhances its transcription activity [14]. YAP/TAZ, in some instances, acts downstream of MEKK3 (or MAP3K3) to maintain stemness in pancreatic cancer cells. Recently, single-cell sequencing data have revealed YAP/TAZ as key regulators of stemness and cell plasticity in glioblastoma [26]. The stem cell transcription factor Sox2 maintains cancer stem cells (CSCs) in osteosarcomas. The Sox2–Hippo axis is conserved in other Sox2-dependent cancers such as glioblastomas. The Sox2–Hippo axis has been shown to directly inhibit Neurofibromatosis Type 2 (NF2) protein and activate YAP, driving cell plasticity in both osteosarcoma and GBM cell lines [27]. Multiple findings have shown that EMT displays spindle cell morphology in cancer pathological tissues. This is an ecological adaptation of cancer cells under interactions of the surrounding microenvironment. In Nasopharyngeal carcinoma, a novel ecological dispersal model of tumor multidirectional progression is proposed, and palatial–EMT has been described in detail elsewhere [28].

YAP/TAZ activity is crucial for liver development, and YAP hyperactivation results in overgrowth. YAP/TAZ activity needs to be greatly balanced to promote healthy organogenesis during development and to avoid post-development tumorigenesis. In recent years, TAZ and YAP fusion proteins have been reported in some cancer types and represent the most common genetic mechanism by which the two transcriptional coactivators are activated. A density-independent nuclear accumulation in osteosarcoma and GBM cell lines for YAP1-TFE3 is suggestive of the refractory nature of the fusion protein to Hippo-mediated inhibition [29].

### 2.2. TEADs Withhold the Oncogenic Driver Seat in Various Types of Tumors

Due to a lack of a DNA binding domain in YAP/TAZ, the activation of the downstream transcriptional program is orchestrated through their interaction primarily with the TEAD family (TEAD1-4) of transcription factors [30]. Although YAP/TAZ lacks apparent druggable pockets, the discovery of the lipid pocket on TEADs has led to the development and evaluation of small-molecule inhibitors (SMIs) targeting TEADs [31,32]. YAP/TAZ and the transcriptional program downstream of the KRAS/MAPK (mitogen-activated protein kinase) pathway have been reported for their cross-talks at multiple levels and converge on an overlapping set of target genes [33,34]. In cell culture conditions, the majority of cancer cells displayed a high distribution of YAP/TAZ in the cytoplasm, and nuclear translocation is significantly increased upon loss of contact inhibition. TEADs serve as the dominant DNA-binding platform for YAP/TAZ. On the genome-wide scale, TEAD consensus motifs are found in most YAP/TAZ-bound cis-regulatory elements at promoters and enhancers [35]. Next to the TEAD, the second most frequent motif at YAP/TAZ-bound peaks corresponded to the consensus for Activator Protein-1 (AP-1) [36]. However, unlike the TEAD transcription factor, the role of AP-1 in the context of YAP/TAZ-driven pathophysiology in mis-regulated Hippo signaling is poorly understood, although YAP/TAZ-TEAD is believed to collaborate regularly with AP1 in transcriptional outcomes. TEADs’ transcriptional output also regulates Hippo–YAP/TAZ signaling via target gene expression. Upon the suppression of Hippo signaling, YAP/TAZ is translocated into the nucleus and interacts with TEADs, thus regulating the expression of the target genes. Upon activation, TEADs are fine-tuned with multiple oncogenic signaling including Wnt, KRAS, and EGFR, fueling the process of carcinogenesis. Imbalanced TEADs transcriptional output leads to the signal enhancement of many oncogenes, including KRAS, BRAF, LKB1, NF2, and MYC, and subsequently facilitates tumor progression, metastasis, cancer metabolism, and immunity, and serving as the hallmark of cancer [37]. Wang and colleagues analyzed a TCGA dataset and reported a 14% (67/478 samples) TEADs alteration in patients with stomach adenocarcinoma. Notably, TEAD3 and TEAD4 amplification in genetic alteration was 30% and 36%, respectively. TEAD4 mRNA was upregulated in patients with GC, and a significant enrichment (~7%) was predicted in gastric intestinal-type adenocarcinoma. Increased TEADs (TEAD1-4) mRNA expressions were significantly correlated with overall survival (OS), progression-free survival (PFS), and post-progression survival (PPS) in gastric cancer (GC) patients [38]. A pan-cancer differential TEADs expression identified a high expression of TEAD1, TEAD2, TEAD3, and TEAD4 in 3, 6, 5, and 12 types of cancer tissues, respectively. Among the cancers with a high TEAD expression, the expression of TEAD4 displayed the highest correlation with the poor prognosis of clear cell renal cell carcinoma (ccRCC). TEAD4 depletion significantly reversed the malignant phenotypes of ccRCC [39]. These findings are in line with earlier data suggesting that TEAD4 is significantly overexpressed in gastric, colorectal, liver, breast, lung, and oral cancers, and the high TEAD4 expression is closely related with tumor development. Since drugging the Hippo pathway is incredibly challenging, druggable hydrophobic pocket in TEADs has emerged as an attractive target.

### 2.3. Targeting TEAD-Driven YAP/TAZ Signaling and YAP Amplification as the Monotherapy

Including YAP/TAZ amplification, upstream Hippo signaling components’ mutations or suppression are linked to multiple cancer types. Clinical data also suggest that YAP/TAZ–TEAD activity are additive to the process of tumorigenesis in a broad range of cancers [19]. Recently, Shamaine and colleagues identified TEAD1 association with mesenchymal-subtype GC (Mes- GC) enhancers, representing a potential therapeutic target for Mes- GCs. In a xenograft model, TEAD1 depletion resulted in a strong tumor regression [40]. Multiple preclinical data generated from cell line-derived xenografts (CDX) showed that TEAD inhibitors, including the recently developed leads (VT3989, IK-930, IAG933, and BPI-460372), are efficacious in YAP/TAZ-dependent cancer types. There are multiple TEAD inhibitors being developed at the academic and industrial research and development (R&D) settings (Figure 1). Multiple good binders that look like L-shaped compounds bind well into the TEAD palmitoylation pocket. However, recent Y-shaped designs strongly occupy a large hydrophobic pocket and potentially enhance the potency [41]. Multiple covalent and non-covalent TEAD binders with good potency are neither L-shaped nor Y-shaped compounds (Figure 2). A structure activity relationship (SAR)-based small-molecule inhibitor design, supported with powerful artificial intelligence (AI), machine learning (ML), and bioinformatics tools, is a great viable option for developing novel TEAD binders as inhibitors for the YAP/TAZ_TEAD complex. Using the TEAD inhibitor for monotherapy has been reviewed recently [42]. Multiple disease indications, compound name, and status are indicated in Table 1.

### 2.4. Oncogenic Driver K-RAS Mutation, Oncogenic Adaptations, and Combination Therapy

KRAS mutations are among the most common genetic alterations in cancer. A total of 25–30% of tumors with KRAS mutations are key drivers in lung, colorectal, and pancreatic cancers. KRAS mutations are found in 32% of lung cancer, 40% of colorectal cancer, and 85% to 90% of pancreatic cancer cases. The most common subtype of lung cancer, non-small-cell lung cancer (NSCLC), is one of the leading causes of death globally [43]. In addition to KRAS amplification, a broad range of allelic mutations including G12A/C/D/F/V/S, G13C/D, V41, L19F, Q22K, D33E, Q61H, K117N, and A146V/T makes it challenging to tackle mutant K-RAS-driven diseases [44]. Among all mutations, G12C is the most common mutation subtype (12–14%) in NSCLC [45]. Two highly potent small-molecule AMG510 (Sotorasib) and MRTX840 (Adagrasib) G12C inhibitors (G12Ci) have made a substantial difference in the field. However, additional co-mutations pose new challenges toward achieving a long-term and sustainable therapeutic outcome. In a large unbiased clinicogenomic analysis of 424 patients with NSCLC, co-alterations of KEAP1, SMARCA4, and CDKN2A were correlated with poor clinical outcomes in patients with KRAS G12C-mutated NSCLC treated either with sotorasib or adagrasib. Misregulated MAPK signaling cascade has also been reported in the development of drug-resistant tumors [46]. A study reported that the adaptive resistance to KRAS G12Ci could be aided by a new synthesis of mutant KRAS in the active state or the activation of the wild-type KRAS isoforms [46]. Additionally, KRAS G12Ci also leads to rapid Receptor Tyrosine Kinase (RTK)-mediated RAS pathway feed-back reactivation, allowing for an escape from the monotherapeutic agents [47]. Taken together, the research findings summarized in this section strongly indicate the possibility of a combination therapy approach to effectively target KRAS-mutant tumors and to address the complexity of co-mutations and the development of drug resistance.

In cancer, YAP/TAZ often gets activated through overexpression or the loss of upstream negative regulators, such as NF2, or non-genetically by a variety of upstream signals [17]. The stability and expression of YAP/TAZ have been shown to be modulated by the RAS family of small GTPases, and the transcriptional program of YAP/TAZ plays pivotal roles in driving oncogenesis [19]. YAP activation can compensate for KRAS inhibition in KRAS-driven murine models of cancer and enable KRAS-independent tumor growth [33,48]. Likewise, YAP/TAZ is known to drive tumor proliferation and resistance in response to a variety of targeted therapies, including EGFR, Anaplastic Lymphoma Kinase (ALK), Mitogen-activated protein kinase (MEK), and Cyclin-dependent kinase 4/6 (CDK4/6) inhibitors [24,49,50,51,52]. For these reasons, pharmacological inhibitors that can block the transcriptional program of YAP/TAZ-TEAD withhold a strong potential in combination cancer therapies.

Recent findings have reported the mechanism of adaptive resistance to KRAS G12C inhibitors in the activation of an EMT program and feedback reactivation of MAPK signaling through the upregulation of the RAS-superfamily protein MRAS. Short-term adagrasib or sotorasib exposure induces the cytoplasmic relocalization of the E-cadherin and Scribble (Scrib) proteins and subsequent adaptive resistance to KRAS G12Ci [53,54]. A study also revealed that activated YAP favors the increased expression of MRAS, which, by promoting the assembly of the MRAS/SHOC2/PP1 complex, bypasses adaptive resistance to G12Ci [55].

In a resistance model, sotorasib retained its ability to alkylate KRAS G12C and suppress the MAPK pathway. A Pan-TEAD inhibitor (GNE-7883) overcomes sotorasib resistance by suppressing the transcription of YAP/TAZ target genes. In an SW837 xenograft model, the tumor continued to grow under sotorasib treatment. In contrast, the sotorasib and GNE-7883 combination led to a robust anti-tumor response and enhanced efficacy compared to the sotorasib single-agent treatment [56].

In a genome-wide CRISPR/Cas9 screening, TEAD inhibition was identified to be synthetically lethal to KRAS G12Ci, credentialing the ability of TEAD inhibition to enhance KRAS G12Ci efficacy [57]. Indeed, the co-occurrence of the TEAD alteration is in line with KRAS alterations across a broad range of tumors. The Pan-TEAD inhibitor, VT-104, in combination with G12Ci, resulted in growth inhibition across multiple cell lines. In combination, the TEAD inhibitor was sufficient to enhance the long-term efficacy of G12Ci. In xenograft and patient-derived xenograft models, the TEAD inhibitor (VT3989) in combination with adagrasib significantly delayed tumor regrowth without causing significant toxicity [58]. A model of combination therapy strategy to target KRAS^G12C^-driven tumor is illustrated in Figure 2. Collectively, the research findings discussed in this section support concurrent TEAD inhibition as a combination therapeutic strategy to improve the long-term efficacy of the pharmacological inhibition of mutant KRAS tumors.

### 2.5. Targeting YAP/TAZ-TEAD Signaling in EGFR Driven Tumors

The interaction of the Hippo pathway with EGFR signaling and HPV oncoproteins in the progression of cervical cancer was reported in 2015. This study indicates a combination therapy strategy of targeting the Hippo and the ERBB signaling pathway for the prevention and treatment of cervical cancer [59]. In response to EGFR-TKIs treatment, the expression of TAZ has been demonstrated as an intrinsic mechanism of T790M-induced resistance. Targeting EGFR and TAZ together may enhance the efficacy of EGFR-TKIs in the acquired resistance of NSCLC [60]. A study also suggests that in EGFR wild-type NSCLC, TAZ-driven amphiregulin (AREG) expression triggers the activation of the EGFR signaling pathway, and Gefitinib induced sensitization in EGFR wild-type NSCLC [61]. YAP and its downstream target AXL play an important role in the development of resistance to EGFR TKIs, suggesting that a combined inhibition of EGFR and the YAP/AXL axis could be a therapeutic strategy for selected NSCLC patients [62]. YAP/TAZ-TEAD downstream transcriptional target gene AXL is one of the known TAM (TYRO3, AXL, and MERTK) family member, an important RTK involved in resistance to EGFR inhibitors [63,64]. AXL heterodimerizes with non-TAM-family RTKs, especially EGFR, in a ligand-dependent or -independent fashion. A recent study suggested that AXL heterodimerizes with EGFR, thereby activating YAP via the EGFR–LATS1/2 axis, a potential reason for head and neck squamous cell carcinoma (HNSCC) [65]. A high level of AXL activation plays an important role in EGFR-driven NSCLC cells treated with the EGFR-TKI Osimertinib and the development of the intrinsic resistance of EGFR-mutated lung cancer. AXL interacts with EGFR and HER3 to maintain the activation status of the downstream signal pathway, conferring intrinsic resistance to osimertinib in NSCLC [66]. Multiple signaling pathways are related to EGFR activation induced by TEAD transcriptional target CTGF and EGFR interaction. CTGF binds to EGFR through its C-terminal module. This interaction activates the EGFR signaling pathway linked to the modulation of different pathways closely related to cell proliferation, inflammation, EMT, and fibrosis in renal damage [67]. TEAD-driven YAP/TAZ transcriptional targets mediate oncogenic adaptations and the development of drug resistance in EGFR-mutant tumors [50]. The blockade of ERK1/2 reactivation by combined EGFR/MEK inhibition uncovers cells that survive by adapting the senescence-like dormant state, characterized by high YAP/TEAD activity. YAP/TAZ-TEAD engages the EMT transcription factor SLUG to directly repress the pro-apoptotic Bcl-2 modifying factor (BMF), inhibiting drug-induced apoptosis. The pharmacological co-inhibition of YAP and TEAD reduces dormant cells by enhancing EGFR/MEK inhibition-induced apoptosis. This study proposes the strategy of co-targeting EGFR, MEK, and YAP/TAZ-TEAD to improve the treatment efficacy in EGFR-mutant NSCLC. A three-drug combination could be an attractive approach; however, toxicity concerns should not be undermined [50]. The inhibition of MEK or EGFR with small molecules drives TEAD transcriptional activity, which is suggestive of a TEAD-pathway upregulation as an adaptative response to oncogene inhibition. The TEAD inhibitor IK-930 resulted in the disruption of a TEAD-driven transcription induced by MEK or EGFR inhibitors. A combination of IK-930 with MEKi and EGFRi resulted in antitumor activity in vivo in KRAS- and EGFR-mutant xenograft models of colorectal carcinoma cells and NSCLC. A series of TEAD inhibitors generated by Vivace therapeutics were evaluated in in vitro and in vivo models of the combination therapy. Indeed, the VT3989 and Osimertinib combination displayed synergy in EGFR-mutant NSCLC [68]. A model of combination therapy strategies to target various forms of EGFR-mutant cancer cells and oncogenic adaptations is illustrated in Figure 3.

## 3. Conclusions and Perspectives

Since the conceptualization of the Hippo signaling pathway in 2007 [69], the core components have been shown to be vital for the maintenance of various cellular processes including cell growth, proliferation, and regenerative functions. 

It is not surprising that the misregulation of the Hippo pathway is linked to an array of pathological consequences. Misregulated Hippo signaling is sufficient to turn the healthy state of cell signaling into a wide range of pathologic conditions and serve as the hallmark of cancer. Cross-talk between Hippo signaling components with multiple oncogenic proteins during the process of carcinogenesis only recently gained a wider attention, leading to the concept that YAP/TAZ-TEAD is the converging point for diverse signaling pathways or the converging point of the intracellular signaling network. Targeting Hippo signaling components in NF2-mutant mesothelioma, gastric cancer cells, neuroblastoma, and any other indications with YAP/TAZ amplification as well as fusions, have emerged as new therapeutic avenues to treat a wide range of cancer patients. Recent findings are very encouraging for developing small-molecule TEAD inhibitors as the combination therapy approach to efficiently target mutant EGFR- and KRAS-driven tumors and the development of drug resistance. In addition to YAP/TAZ-TEAD hyperactivation, their transcriptional target genes (e.g., AXL, CTGF, CYR61) also play a key role in the oncogenic adaptations and development of drug resistance. Although we are inching closer to clinical trials with multiple TEAD inhibitors, more questions still need to be answered regarding the nature of TEAD binders (central pocket vs. Interface 3) and drug-induced toxicity in the long run: what are the predictable biomarkers for stratifying cancer patients? What indications are most suitable for the YAP/TAZ-TEAD inhibitors? Additional questions that we need to address include the following: do we need to target all four TEADs, and what are the functional redundancies among TEAD1-4 during the process of tumorigenesis? Unlike cytoplasmic and membrane-bound proteins, the complexity of drugging chromatin-bound TEAD1-4 via targeted protein degradation (TPD) is still challenging. Moreover, the identification of novel TEAD regulators to downregulate nuclear TEAD and subsequent YAP/TAZ association to halt the carcinogenesis could be an interesting approach.

Collectively, the role of Hippo signaling components and their cross-talk with mutant KRAS and EGFR proteins during the process of carcinogenesis, oncogenic adaptation, and development of drug resistance are in line with the concept that YAP/TAZ-TEAD is the converging point of the cellular signaling network regulating cell proliferation and apoptosis, and targeting YAP-TAZ-TEAD is a viable approach to monotherapy and combination therapy to offer novel treatment options for cancer patients in the coming years.

## Figures and Tables

**Figure 1 cells-13-00564-f001:**
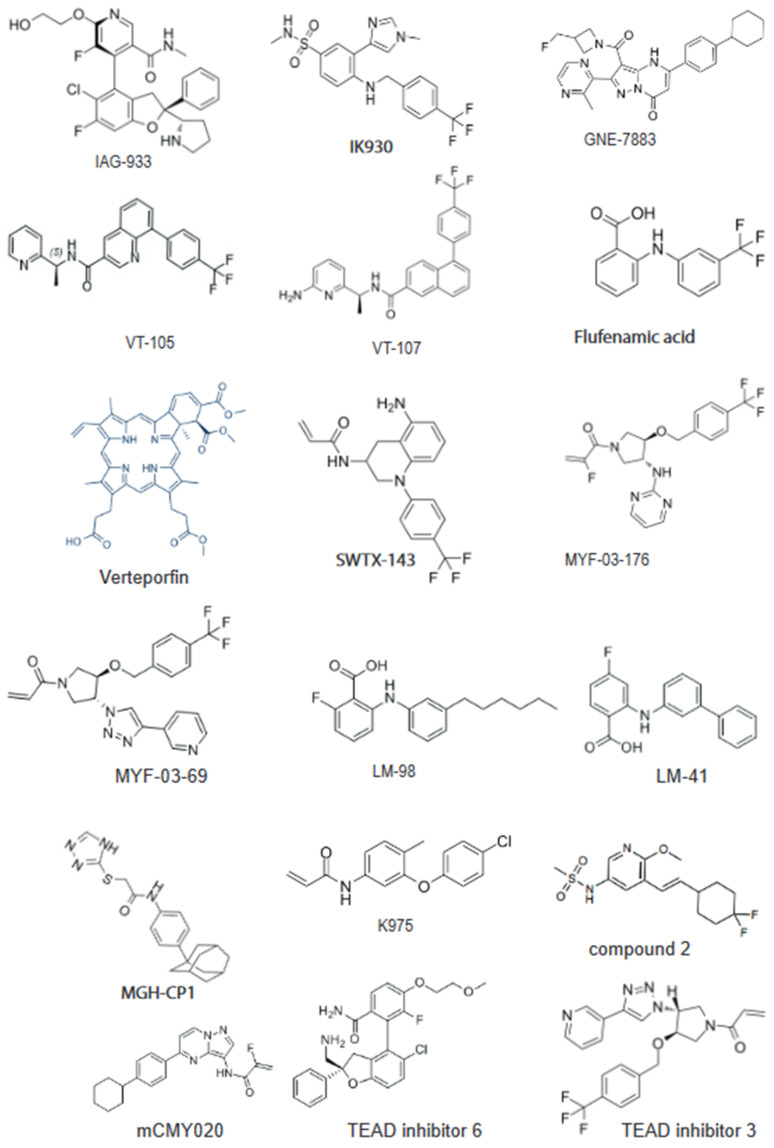
Chemical structure of key TEAD inhibitors: IAG933 and the inhibitor 6 bind to the surface of TEAD at interface 3. Majority of inhibitors are designed against the central hydrophobic pocket. Lead molecules IK-930, VT107, GNE-7883, SWTX-143, Flufenamic acid (FA), and MGH-CP1 are non-covalent binders. K-975, mCMY020, MYF-03-176, and MYF-03-69 are covalent inhibitors targeting conserved cysteine residues in the pocket.

**Figure 2 cells-13-00564-f002:**
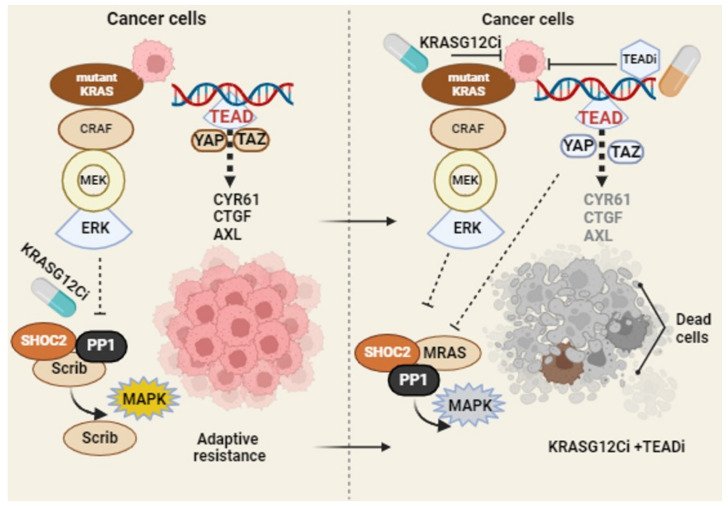
Targeting adaptive resistance of KRAS G12C mutation: in mutant KRAS cancer cells, KRAS inhibitors trigger mis-localization of Scrib and nuclear localization of YAP. YAP/TAZ-TEAD signaling and MAPK reactivation fuel the process of adaptive resistance in the presence of KRAS G12Ci (left panel). The combination of small-molecule Pan-TEAD inhibitor impairs the YAP/TAZ-TEAD nuclear binding and expression of transcriptional target genes, limiting adaptive resistance to KRAS inhibitors. Created in BioRender.com (accessed on 27 February 2024).

**Figure 3 cells-13-00564-f003:**
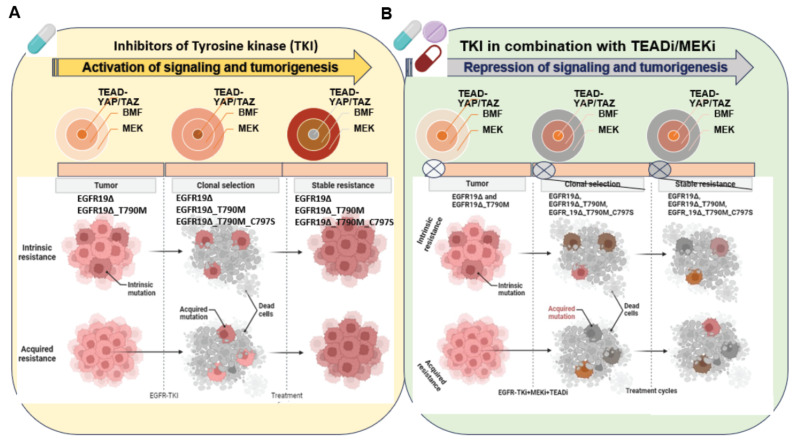
Strategy to target tumors with mutant EGFR and the development of drug resistance: in NSCLC, various forms of EGFR mutations led to intrinsic and acquired resistance in response to a tyrosine kinase inhibitor treatment. YAP/TAZ-TEAD engages the EMT transcription factor SLUG to directly repress pro-apoptotic BMF, limiting drug-induced apoptosis (**A**). The combination therapy of MEK/TEAD inhibitors disrupts the process of tumor evolution and development of drug resistance (**B**). Note: this model also covers a broad range of other EGFR mutations including EGFR exon 20 insertions and exon 21 mutations. Created in BioRender.com (accessed on 27 February 2022).

**Table 1 cells-13-00564-t001:** Current progress with TEAD inhibitors, including Phase 1 molecules.

Compounds	Company	Target	Status
**VT3989**	Vivace Therapeutics	Pan-TEAD	**Ph1 (NCT04665206):** NF2-mutated tumors, advanced pleural malignant mesothelioma or other metastatic solid tumors, and patients with mutations of NF2 that have progressed on or following standard therapy.
**IK-930**	Ikena Oncology	TEAD1	**Ph1 (NCT05228015):** Mesothelioma, NF2 deficiency, other NF2-deficient solid tumors, and solid tumors with YAP1/TAZ genes fusion
**IAG933**	Novartis	YAP-TEAD	**Ph1 (NCT04857372):** mesothelioma, NF2 mutant, and Yap/Taz fusion tumors, malignant pleural mesothelioma, NF2 truncating mutations or deletions, solid tumors with functional YAP/TAZ fusions, and NF2/LATS1/2 mutated tumors.
**BPI-460372**	Betta Pharma	Pan-TEAD	**Ph1 (NCT05789602):** selective TEAD palmitoylation inhibitor solid tumor study in China
**ODM-212**	Orion	Pan-TEAD	**Ph1** (NA) solid tumors with YAP/TEAD activation
**TY-0584**	Tyk Medicines	YAP-TEAD	IND-enabling
**ETS-003**	Etern Biopharma	YAP-TEAD	Preclinical/IND-enabling
**GH658**	Suzhou Genhouse	Pan-TEAD	Preclinical/IND-enabling
**SW-682**	Springworks	Pan-TEAD	Preclinical/IND enabling
**BGI-9004**	Bridgene Biosciences	Pan-TEAD	Preclinical
**SPR1-0117**	Sporos Biodiscovery	TEAD1, TEAD4	Preclinical
**K-975**	Sanofi/ Kyowa Kirin	YAP-TEAD	Preclinical
**SJP1901**	Samjin Pharmaceutical	Pan-TEAD	Preclinical
**KYP-1104**	Samjin Pharmaceutical	YAP-TEAD	Preclinical
**GNE-7883**	Genentech	YAP-TEAD	Preclinical

Source: Publicly available information, AACR meeting 2023 and https://clinicaltrials.gov (accessed on 1 March 2024).

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
