# Peer review of "Hippo Signaling at the Hallmarks of Cancer and Drug Resistance"

_cells, 2024, doi:10.3390/cells13070564_

Round 1

Reviewer 1 Report

Comments and Suggestions for Authors

In this review, the authors have highlighted the emerging aspects of hippo signaling cross-talk with other oncogenic divers, and how this information can be translated into combination therapy. Some points should be noted as below,

1) The title“Hippo signaling at the Hallmarks of cancer and development of drug resistance.”, so what is hallmarks of cancer”related to kippo signaling”,it's best to specifically describe it.

2) “YAP/TAZ activation in EMT and Oncogenic stemness”,the description of EMT is too simplistic,how about palatial-EMT?

Many studies have shown that EMT shows spindle cell morphology in cancer pathological tissues. Which is indeed an ecological adaptation of cancer cells under interactions of cancer cells and the surrounding microenvironment(https://www.thno.org/v13p1607.htm). It should be helpful for understating more about EMT.

3) What are the specific tumor tpye that these compounds are used for Table 1? Which cancer hallmark/or pathway is mainly targeted?

4) Please show some future perspectives.

Author Response

In this review, the authors have highlighted the emerging aspects of hippo signaling cross-talk with other oncogenic divers, and how this information can be translated into combination therapy.

Thanks for this reviewer for valuable comments. We have tried our best to address his concerns and improve the manuscript quality as mentioned below:

  • The title“Hippo signaling at the Hallmarks of cancer and development of drug resistance.”, so what is “hallmarks of cancer”related to “kippo signaling”,it's best to specifically describe it.

TEAD driven transcriptional programme and it’s pairing with YAP/TAZ signaling is the Hallmark of cancer.

  • “YAP/TAZ activation in EMT and Oncogenic stemness”,the description of EMT is too simplistic,how about palatial-EMT?

Many studies have shown that EMT shows spindle cell morphology in cancer pathological tissues. Which is indeed an ecological adaptation of cancer cells under interactions of cancer cells and the surrounding microenvironment(https://www.thno.org/v13p1607.htm). It should be helpful for understating more about EMT.

Thanks reviewer to mentioning this point. In this review we have focused on the role of hippo signaling components in the EMT. Thank for mentioning key point regarding palatial-EMT.

As you have suggested in the comment, we have included key point in this section.

  • What are the specific tumor tpye that these compounds are used for Table 1? Which cancer hallmark/or pathway is mainly targeted?

Thanks for this suggestion. We have added more detail regarding tumor/cancer types. All these compounds are directed against Hippo signaling pathway and targeting YAP/TAZ and TEAD signaling.

Please show some future perspectives.

We have added conclusion and future perspective at the end of this review.

Reviewer 2 Report

Comments and Suggestions for Authors

The manuscript "Hippo signaling at the Hallmarks of cancer and development of drug resistance" presents interesting data but requires several modifications before being considered for publication.

- The manuscript needs to be revised extensively in order to improve fluency. In its current state, the manuscript has several very choppy parts that make it difficult to read and understand.

- All abbreviations must be defined the first time they appear in the manuscript (e.g. MOB1A/B, TEADs, etc.)

- At the end of the introduction, the purpose of the manuscript and its contribution to the scientific community should be described.

- Given the nature and impact of the journal, you should delve deeper into the topics. In its current state it is merely a summary that does not provide an important contribution to the scientific community. There are already many similar articles in the literature.

- The quality of the figures needs to be improved.

- You should add a section of conclusions as well as future perspectives.

- I don't understand why Imatinib is mentioned in the abstract but never mentioned in the body of the manuscript. I can't find the connection with Hippo signaling

- Disclosure sections are missing (conflict of interest, authors' contribution, etc.)

- Where TAZ is mentioned the first time it should be modified to:...its paralog TAZ (also known as WW domain containing transcription regulator 1 [WWTR1]).

- The role of Hippo signaling in the cell cycle, ferroptosis and antitumor immunity should also be discussed.

Comments on the Quality of English Language

The manuscript needs to be revised extensively in order to improve fluency. In its current state, the manuscript has several very choppy parts that make it difficult to read and understand the manuscript.

Author Response

The manuscript "Hippo signaling at the Hallmarks of cancer and development of drug resistance" presents interesting data but requires several modifications before being considered for publication. The manuscript needs to be revised extensively in order to improve fluency. In its current state, the manuscript has several very choppy parts that make it difficult to read and understand.

Thank you very much for providing valuable comments and suggestion. 

 We have revised the manuscript to edited adequately to improve the flow.

- All abbreviations must be defined the first time they appear in the manuscript (e.g. MOB1A/B, TEADs, etc.)

We have corrected abbreviations and now well defined.

- At the end of the introduction, the purpose of the manuscript and its contribution to the scientific community should be described.

Thanks for bringing this key point. We have added conclusion and perspective section at the end which covers your suggestions.

- Given the nature and impact of the journal, you should delve deeper into the topics. In its current state it is merely a summary that does not provide an important contribution to the scientific community. There are already many similar articles in the literature.

Thanks for this suggestion, we have worked in it.

- The quality of the figures needs to be improved.

Thanks for noticing this. We have worked and improved figures now. 

- You should add a section of conclusions as well as future perspectives.

-Thanks for bringing this key point. We have added it.

- I don't understand why Imatinib is mentioned in the abstract but never mentioned in the body of the manuscript. I can't find the connection with Hippo signaling.

We have mentioned Imatinab to give first example of the clinical Tyrosine Kinase inhibitor. Our objective is to highlight why do we need alternative combination therapy to target mutant specific EGFR tumors.

- Disclosure sections are missing (conflict of interest, authors' contribution, etc.)

We have made disclosure on conflict of interest in the manuscript submission section.

- Where TAZ is mentioned the first time it should be modified to:...its paralog TAZ (also known as WW domain containing transcription regulator 1 [WWTR1]).

Thanks, we have made this change.

- The role of Hippo signaling in the cell cycle, ferroptosis and antitumor immunity should also be discussed.

Due to word limitations, we have primarily focused on carcinogenesis and development of drug resistance.

The manuscript needs to be revised extensively in order to improve fluency. In its current state, the manuscript has several very choppy parts that make it difficult to read and understand the manuscript.

We have worked extensively and edited extensively for a better reading flow.

Reviewer 3 Report

Comments and Suggestions for Authors

In this manuscript, authors summarized the major components of canonical Hippo pathway and discussed various mechanisms about dysregulation of Hippo pathway. Authors further discussed current developments in small molecules that targeting YAP/TAZ and TEAD transcription factors as well as potentially combination therapy targeting RAS and EGFR signaling.

Overall, the manuscript is well written and informative.

One or two paragraphs discuss about further directions and summary are highly recommended.

Author Response

In this manuscript, authors summarized the major components of canonical Hippo pathway and discussed various mechanisms about dysregulation of Hippo pathway. Authors further discussed current developments in small molecules that targeting YAP/TAZ and TEAD transcription factors as well as potentially combination therapy targeting RAS and EGFR signaling.

Overall, the manuscript is well written and informative.

Thanks to this reviewer for encouraging comments and valuable comments. We have worked accordingly.

One or two paragraphs discuss about further directions and summary are highly recommended.

We have added conclusion and future direction.

Reviewer 4 Report

Comments and Suggestions for Authors

Dear Author,

In this review, you have emphasized the emerging aspects of Hippo signaling cross-talk with other oncogenic pathways and discuss how this knowledge can be translated into more effective combination therapies for cancer treatment. Despite significant strides in cancer therapy, including the approval of nearly 100 small molecule anti-cancer drugs by regulatory authorities such as the FDA and China NMPA since the introduction of imatinib in 2001, challenges remain. Low response rates and the emergence of drug resistance pose substantial obstacles to improving progression-free survival (PFS) and overall survival (OS) in cancer patients.

Please follow the instruction from the editor.

Comments on the Quality of English Language

Minor editing required.

Author Response

  1. The author highlighted few compounds in Table 1 is from the AACR meeting 2023, which is not providing the enough review about hippo signaling inhibitor. The author should consider from the clinical studies around the world (https://clinicaltrials.gov/).

Thank you very much for encouraging comments and valuable suggestions. I am very happy to work on your suggestions

We have used the link and updated clinical trial numbers searched through the link.

  1. The author forgot to acknowledge Biorender in Figure 2.

 Thanks for pointing this. We have added now.

  1. The author did not mention the efficacy and potential limitations of targeting TEAD-driven YAP/TAZ signaling as a monotherapy in cancer treatment?"

We have only opportunity in selected tumors like NF2 mutant mesothelioma or high TEAD expressing tumors. Having Biomarker is still challenging. So, I have highlighted these challenges in the conclusion and future perspectives.

  1. What are the therapeutic implications and how does targeting the YAP/TAZ-TEAD signaling complex impact EGFR-driven tumors?

In mutant EGFR there are high expression of AXL and CYR61 (TEAD transcriptional target genes) which play key role in development of drug resistance. So, targeting TEAD arm serve as the potential therapeutic opportunity.

Kindly refer to the section “Targeting YAP/TAZ-TEAD signaling in EGFR driven tumors” I have highlighted key points in this section.

  1. How can the interplay between oncogenic K-RAS mutations and the YAP/TAZ signaling pathway drive oncogenic adaptations in cancer, and what are the implications for combination therapy?

Thanks for asking this great point.

YAP/TAZ targets upregulated by oncogenic RAS and mutant specific adaptations are poorly understood. In this review we have highlighted drug induced interplay between oncogenic K-RAS mutations and the YAP/TAZ signaling pathway drive oncogenic adaptations in cancer. In the figure we have tried to illustrate it.

Thank you.

Round 2

Reviewer 1 Report

Comments and Suggestions for Authors

No other questions

Author Response

Thanks to the reviewer.

Reviewer 2 Report

Comments and Suggestions for Authors

The manuscript has undergone improvements compared to the previous version. I would suggest enhancing the quality of the figures, or at the very least, removing the Biorender labeling for a cleaner presentation.

Comments on the Quality of English Language

Moderate editing of English language required

Author Response

Thank you very much for your valuable comments. We have worked on your suggestions and edited to enhancing the quality of the figures including removing the Biorender labeling for a cleaner presentation.

We look forward for your kind consideration.

Best Regards, 

Ramesh Kumar